# Exploring the Spatial Image of Traditional Villages from the Tourists' Hand-Drawn Sketches

**Zuoming Jiang [1,2] and Yang Sun [3,*]**

1   School of History, Culture and Tourism, Huaibei Normal University, Huaibei 235000, China; jiang21@stu.xmu.edu.cn
2   School of Management, Xiamen University, Xiamen 361005, China
3   School of Fine Arts, Huaibei Normal University, Huaibei 235000, China
*   Correspondence: syzm586@163.com

**Abstract:** As an important concept in cognitive psychology and behavioural geography, destination spatial image cognition has a significant impact on the quality of tourists' experience, and on their behavioural intention. However, studies of spatial image cognition in small-scale traditional villages are limited. Therefore, the present study analyses the spatial image characteristics of four traditional villages of World Cultural Heritage sites in China through the use of tourists' hand-drawn sketches, using a sample of 366 respondents to further explore the evolution process of cognitive map types and constituent elements with tourists' stay days. Results indicate that the spatial cognitive map and landmarks are the main types and dominant elements of spatial image cognition, respectively. The tourists' spatial cognitive process includes two sequences, as follows: the evolution sequence of dominant cognitive maps is "spatial + individual → spatial + individual + hybrid → spatial + individual", while the evolution sequence of dominant cognition elements is "landmark + path + animal and plant → landmark + animal and plant + path". This study extends the current destination spatial image cognition literature, and has substantial value for the destination in terms of developing traditional village sustainable tourism based on the tourists' attitude, as obtained by the cognitive map method.

**Keywords:** tourism destination image; spatial cognitive process; cognitive map; traditional villages

## 1. Introduction

Spatial cognition originating from the cognitive map has been a hot topic in cognitive psychology and behavioural geography. In this field, the most foundational research is Lynch's book, *Urban Image* [1]. He concluded that the characteristic elements of urban spatial image include paths, edges, districts, nodes, and landmarks. He pioneered the empirical research method of exploring spatial cognition by using hand-drawn cognitive sketches. On this basis, Appleyard divided the types of cognitive map and proposed the hypothesis of the development process of the cognitive map, that is, with the deepening of the familiarity with the city, people's cognitive map developed from a sequential type to a spatial type [2]. Subsequently, many studies considered the five elements of spatial image proposed by Lynch and Appleyard's cognitive map classification method as the basis for exploring people's spatial image cognition rules [3]. However, scholars' discussions on spatial cognition are far from reaching a consistent conclusion. The focus of debate lies in the dominant elements of spatial cognition and the evolution rules of cognitive map types. The former mainly includes three different theoretical viewpoints of path dominance [4], landmark dominance [5], and integrated development [6], whereas the latter has been questioned and debated by empirical studies in other cities since the birth of the Appleyard hypothesis [7–9]. Therefore, empirical research on different groups and different types of cases is very important for solving disputes and promoting further development of spatial cognition theory.

In recent years, the research on spatial image of tourism destinations has had great importance attached to it by scholars [10–14], and the exploration of tourists' spatial cognitive pattern has been gradually deepened [15,16]. However, the research on spatial images of tourism destinations is deeply influenced by urban image research [17], showing the development trend of focusing remarkably on spatial structural analysis of the destination public cognitive map [18], whereas the research results of destination spatial cognitive process are still relatively weak. Guy et al. [19] conducted a study on Wurzburg, Germany, which showed that, compared with indirect information sources, such as path signs, maps, and tourism brochures, onsite experience has the most profound impact on the spatial cognition process of tourists. Walmsley et al. [20] found that in a study of Coffs Harbour, Australia, spatial cognitive elements of tourists showed a trend of fluctuating development over time, and that the cognitive map evolved from a spatial type to a sequential type. Erem [21] proposed a dual-channel cognitive mapping model with personal and environmental variables, and revealed the recreational settlement image of a Mediterranean holiday village from tourist sketch maps. Therefore, tourists' destination spatial cognition process has a unique law. Extending the research horizon from traditional urban residents to tourists can provide new empirical evidence for the study of the spatial cognition process and promote the improvement of the theory of spatial cognition development process with more universal significance.

At present, scholars mostly focus on large-scale urban tourism destinations as the study area of tourists' spatial image cognition. However, exploration for traditional villages, a destination with small spatial scale but abundant tourism attractions, is limited. To fill the research gap of small-scale destination spatial cognition in traditional villages, this study attempts to explore the following questions which remain largely unanswered: (1) What are the dominant elements of spatial image cognition of tourists in traditional village destinations? (2) Is it landmark dominant or path dominant? (3) What is the evolution process of spatial image cognition of tourists in traditional villages? (4) Does it show a pattern of transformation from sequential type to spatial type, consistent with urban destinations? In view of these research questions, this study summarises the dominant elements and map types of spatial image cognition from traditional village sketch hand-drawn by tourists. Furthermore, through statistical analysis of cross-linked tables and graphical display, the evolution trend of spatial cognitive dominant elements and map types with tourists' stay time is studied. By studying these problems, we hope to promote the development of tourists' spatial image cognition in depth.

## 2. Literature Review

According to the existing literature, destination spatial image cognition mainly focuses on the dominant elements of spatial cognition and cognitive map types.

### 2.1. Dominant Elements of Spatial Cognition

Previous studies have shown that people's cognition of unfamiliar environments is a gradual development process. As time goes by, people's cognition of a spatial environment is gradually deepened, and then a systematic cognitive representation is formed in the brain [1,2]. This cognitive map is not only a direct reflection of the inner cognitive representation of spatial environmental information, but also a reflection of the psychological representation of the personal spatial environment, so it is used to explore the richness of personal spatial knowledge and spatial cognitive process [22–24]. The dominant elements of a cognitive map are important references for people to locate in the geographical environment. According to the spatial knowledge highlighted by a cognitive map, the map can be divided into a route map and a survey map [25]. Siegel et al. [5] proposed a relatively complete type of spatial knowledge. They divided spatial knowledge into the following three types: landmark, route, and survey, according to the development level from low to high. The spatial cognitive maps proposed by subsequent scholars are influenced by this classification idea to varying degrees [26]. However, by judging the dominant elements

highlighted by the cognitive map and its development trend, scholars have debated the process of personal spatial cognition. The three main viewpoints in the debate are path dominance, landmark dominance, and integrated development. Based on Piaget's theory of stages of cognitive development, scholars of path dominance believe that the spatial cognitive process is based on the following path elements: topological spatial knowledge was formed initially, and then projective spatial knowledge of locating landmark, and Euclidean spatial knowledge of describing the whole region was developed [4,27]. An alternative proposed pathway is the dominant element of spatial cognition. Siegel et al. [5] believed that the sequence of spatial cognition begins with a landmark. They put forward a three-stage cognitive theoretical model from landmark to path, and finally formed survey space. Kitchin [6] believed that spatial cognitive elements do not strictly hierarchically evolve but are interspersed with each other. Thus, he put forward the theoretical hypothesis of integrated development.

### 2.2. Cognitive Map Types

Based on Lynch's five elements of spatial image, Appleyard [2] proposed a detailed classification of the cognitive map, which has become one of the most commonly used methods for spatial cognition research [18]. According to Appleyard's empirical research in Guyana [2], cognitive maps are divided into the following two categories: sequential and spatial. Sequential maps are dominated by paths and nodes, and are divided into four subcategories of fragmented, chain, branch/loop, and network. On the other hand, spatial maps are dominated by landmarks and districts, and are divided into four subcategories of scattered, mosaic, linked, and patterned. Subsequently, many scholars found new types of cognitive maps on the basis of this classification, among which the most typical are hybrid and individual. Huynh et al. [28] found cognitive maps with significant path and landmark elements and classified them as a hybrid map type. Song et al. [29] classified cognitive maps that reflect path topological relations but lack recognisability of district spatial elements as hybrid types. Bomfim et al. [30] added individual type categories when exploring the emotional dimension of a cognitive map, which only included landmark landscapes but were more abstract and symbolic. Young [31] found that some cognitive maps basically have no spatial representation, but only consist of some landmark landscapes, such as trees and waterfalls. He referred to such cognitive maps as the individual type, and believed that they revealed the image of tourism destinations and could promote in-depth study of tourists' spatial cognition. Table 1 summarises the existing literature on cognitive map types.

In summary, the debate on the spatial cognition process is still ongoing, and the research focus is mostly on local residents. Recent studies have preliminarily revealed the characteristics of spatial cognition of tourists as being different from local residents. For example, cognitive maps are simpler and contain fewer spatial environment features [11,20], and a smaller gap is found in the proportion of sequential and spatial cognitive maps [13]. However, most cases studied are urban destinations, and the spatial cognitive map specifically for traditional villages has not yet been found. Compared with urban destinations, the spatial scale of traditional villages is smaller, but the spatial environmental elements are more centralised. As a result, the types and dominant elements of cognitive maps drawn by tourists in cities and traditional villages may have significant differences. Therefore, this study considers four traditional villages in China's World Cultural Heritage sites as research cases to explore the patterns and characteristics of tourists' cognition of the spatial image of traditional villages.

**Table 1.** Research literature on cognitive map types.

| Author | Study Area (Nation) | Research Object | Cognitive Map Type | | | | |
|---|---|---|---|---|---|---|---|
| | | | Sequential | Spatial | Hybrid | Individual | Others |
| Appleyard [2] | Guyana (Venezuela) | resident | √ | √ | | | |
| Huynh et al. [28] | Toronto (Canada) | resident | √ | √ | √ | | |
| Bomfim et al. [30] | Barcelona and São Paulo (Spain and Brazil) | resident | √ | √ | | √ | |
| Pocock [32] | Durham (United Kingdom) | resident | √ | √ | | | |
| Wong [33] | Hong Kong (China) | resident | √ | √ | | | |
| Feng [34] | Beijing (China) | resident | √ | √ | | √ | |
| Zhang et al. [35] | Lanzhou (China) | resident | √ | √ | | √ | |
| Uusitalo [11] | Lapland (Finland) | foreign tourist; domestic tourist | √ | √ | | | |
| Tian and Sha [13] | Nanchang (China) | domestic tourist | √ | √ | | | |
| Guy et al. [19] | Wurzburg (Germany) | domestic tourist | √ | √ | | | |
| Walmsley et al. [20] | Coffs harbour (Australia) | domestic tourist | √ | √ | | | |
| Erem [21] | Mediterranean holiday Village (Turkey) | foreign tourist; domestic tourist | | √ | √ | | |
| Young [31] | Daintree and Cape Tribulation (Australia) | domestic tourist | √ | √ | | √ | |
| Lee et al. [36] | Macau (China) | foreign tourist; domestic tourist | | √ | | | |
| Zhang et al. [37] | Xi'an (China) | foreign tourist | √ | √ | √ | √ | |
| Qian and Su [38] | Suzhou (China) | domestic tourist | √ | √ | | √ | √ |

Note: "√" indicates that this type exists.

## 3. Methodology

### 3.1. Case of Traditional Village

Chinese traditional villages have preserved precious cultural heritage of agriculture, as well as acting as the spiritual home of modern people [39,40]. In this study, traditional villages in southern Anhui and Tulou in Fujian were selected as case sites. The field survey sites are distributed in the Xidi and Hongcun villages of Huangshan City, Anhui Province, the Tianluokeng village of Zhangzhou City, and the Hongkeng village of Longyan City, Fujian Province. Traditional villages in southern Anhui are typical of Hui-style architecture and Huizhou culture, while Tulou in Fujian is typical of Tulou architecture and the Hakka culture (see Figure 1). Traditional villages are rich in material cultural heritages, such as dwellings, ancestral halls, ancient trees and bridges, and intangible cultural heritages, such as craft carving, architectural techniques, and folk culture. After more than 20 years of cultural relic protection and tourism development, each traditional village received millions of tourists every year before the COVID-19 pandemic, making this one of the most popular traditional village tourism destinations in China [41]. The number of tourists received in 2019 was 1.22 million at Xidi, 2.76 million at Hongcun, 2.54 million at Tianluokeng, and 3.43 million at Hongkeng (this data comes from the statistics of the tourism government department in the locations of traditional villages in 2019). The number of inns and hotels providing accommodation units for tourists in 2020 was 90 in Xidi, 284 in Hongcun, 22 in Tianluokeng, and 60 in Hongkeng (this data comes from the statistical data of industry and commerce government departments in the locations of traditional villages in 2020). Therefore, the selection of the case sites in this study is typical.

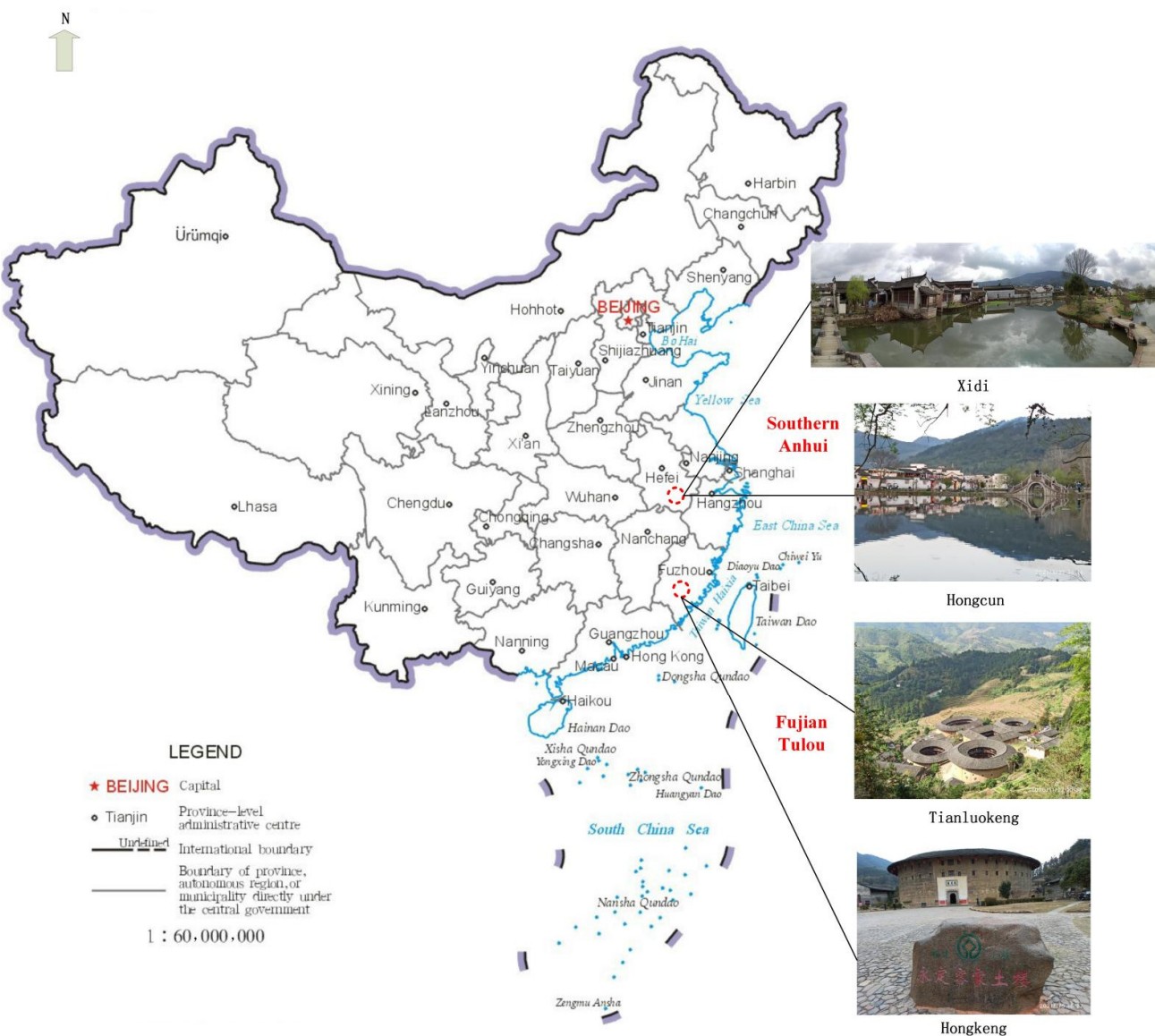

**Figure 1.** The traditional villages in Southern Anhui and Fujian Tulou.

### 3.2. Study Design

This study adopts the questionnaire survey commonly used by scholars to obtain research data. The tourist questionnaire includes two parts. In the first part, tourists are invited to draw a sketch map of traditional villages according to their personal travel experience in the four traditional villages. Although previous studies provided hints of what tourists were expected to draw [19,42], such as examples of hand-drawn sketches or typical markers, this study aimed to avoid the practice effect caused by mutual hints between instructions and sketching tasks [18]. According to Young's practice [31], this study does not provide such hints and gives tourists the maximum degree of freedom and imagination space to fully explore the information on the types and elements of tourists' cognitive map. The second part deals with the demographics and length of stay of tourists, including gender, occupation, age, education level, and monthly income.

### 3.3. Data Collection

The tour routes of group tourism are relatively fixed, and tourists' spatial cognition of traditional villages is often limited to a few scenic spots on the route. This study avoids such groups and conducts a survey for individual tourists. To ensure that tourists had

a deep cognition of traditional villages, qualified respondents had to have spent at least one night in the traditional village or be on at least their second visit to the village. The survey was conducted between 11–23 July and 16–29 August 2021. First, one leisure square and one representative dwelling were randomly selected from each traditional village using the probability-proportional-to-size sampling method. A total of four leisure squares and four representative dwellings constituted sampling units. Second, we used the systematic sampling method to select 60 tourists in each leisure square and each representative dwelling, respectively. Then, 1 tourist was selected from every 20 tourists in the field sampling to participate in this study. A total of 480 questionnaires were sent out, although 114 invalid questionnaires were eliminated because tourists did not draw a sketch map of traditional villages, so 366 valid questionnaires were collected, with an effective rate of 76.25%, including 124 from Tianluokeng, 76 from Hongkeng, 48 from Xidi, and 118 from Hongcun.

Amongst the surveyed tourists, females (54.7%) slightly outnumbered males (45.3%). The top four occupations were manager of enterprise (28.2%), other occupations (17.9%), freelancers (17.1%), and students (17.1%). In terms of age group, the survey was dominated by young people, accounting for 50.8% between 18 and 35, followed by 17.9% between 36 and 45. Most of them had an undergraduate degree (53.8%), followed by postgraduate or above (24.8%), junior college (17.1%), senior high school (2.6%), and junior high school or below (1.7%). The top 2 monthly incomes were more than ¥10,000 (39.3%) and ¥5001–7000 (21.4%), followed by less than ¥3000 (17.1%), ¥3001–5000 (11.1%), and ¥7001–10,000 (11.1%). On 23 July 2021, the exchange rate of CNY against USD is 1:0.15459 and against EURO 1:0.1313. The length of stay was mainly two days (41.0%), followed by one day (35.9%), three days (17.1%), and four days and above (6.0%).

## 4. Results

### 4.1. Cognitive Map Type Analysis

Valid coded questionnaires were distributed to four doctoral students majoring in tourism management for cognitive map type confirmation to improve the rationality of the cognitive map classification. The map types with different classification results were determined through a second round of group discussion. Statistics showed that in the cognitive map of the traditional village spatial image, the proportion of spatial type was the highest, at up to 59.02%. Individual was the second highest, accounting for 32.79%, while cognitive maps integrating landmark and path were classified as hybrid, accounting for 5.46%. The lowest was the sequential type, with a proportion of 2.73%. In general, a traditional village spatial image cognitive map includes the following four types: spatial, individual, hybrid and sequential.

#### 4.1.1. Spatial Cognitive Map

A total of 216 spatial cognitive maps of traditional villages are divided into three subcategories in this study.

(1) Patterned cognitive map (see Figure 2a). This type of cognitive map is dominated by landmarks and districts, mostly centred on traditional village residential buildings, with surrounding mountains and rivers as the regional scope, forming the living space pattern of poetic dwelling in traditional villages. A total of 196 patterned cognitive maps are available, accounting for 53.55% of the total samples.

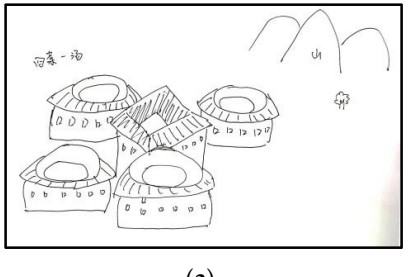 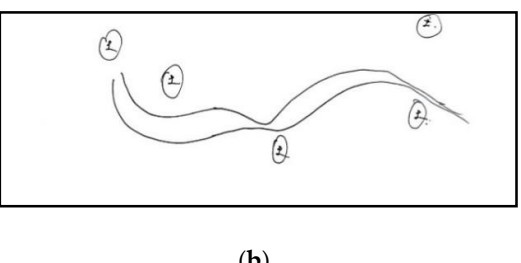 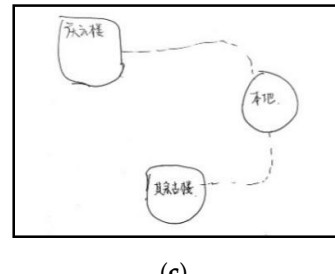

(**a**)  (**b**)  (**c**)

**Figure 2.** Subcategories of spatial cognitive maps of tourists in traditional villages: (**a**) patterned, (**b**) scattered, and (**c**) linked. Note: The Chinese letters in (**a,c**) are the names of Tulou buildings, which are "four dishes and one soup", Qingyun building, and other Tulou buildings, respectively.

(2)   Scattered cognitive map (see Figure 2b). This type of cognitive map does not have a clear district concept, and most of them represent the spatial distribution of traditional village residential buildings in the form of points, circles, and icons. Only 12 scattered-point cognitive maps are available, accounting for 3.28% of the total samples, and most of these were multipoint nonzonal, whereas other district single-point, multipoint, and non-zonal cognitive maps rarely appeared.

(3)   Linked cognitive map (see Figure 2c). This type of cognitive map is mostly presented in the form of path linking residential buildings to outline the tour routes of traditional villages. Only 8 linked cognitive maps are available, accounting for 2.19% of the total samples.

4.1.2. Individual Cognitive Map

The 120 individual cognitive maps in the traditional village spatial image are divided into three subcategories in this study.

(1)   Entity cognitive map (see Figure 3a). This type of cognitive map is presented in the form of realistic landmark landscapes, most of which are beautifully and vividly drawn with an emphasis on landscape details that truly reflect tourists' understanding of the cultural connotations of Fujian Tulou, Hui-style dwellings, and their architectural techniques. A total of 100 entity cognitive maps are available, accounting for 27.32% of the total sample.

(2)   Abstract cognitive map (see Figure 3b). This type of cognitive map is not a realistic representation of the landmark landscape of traditional villages, but a fictionalised depiction of the landmark landscape and its surrounding scenery, or an abstraction of the landmark landscape into some symbols to express. A total of 16 abstract cognitive maps are available, accounting for 4.37% of the total sample.

(3)   Scene cognitive map (see Figure 3c). Such cognitive maps integrate characters into the iconic landscape background of traditional villages, reflect specific tourism scenes, and express the good memories of tourists. The number of scene cognitive maps is very small. Only 4 pictures of tourists taking photos in front of the Tulou and family travel are found, accounting for 1.09% of the total sample, but they vividly show the interaction screen between tourists and traditional villages.

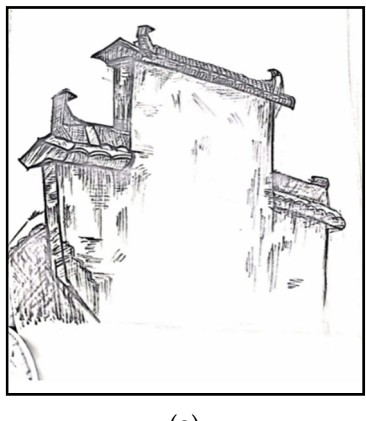 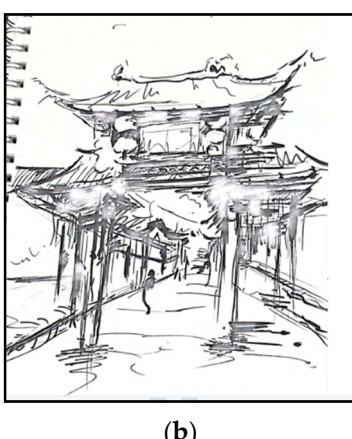 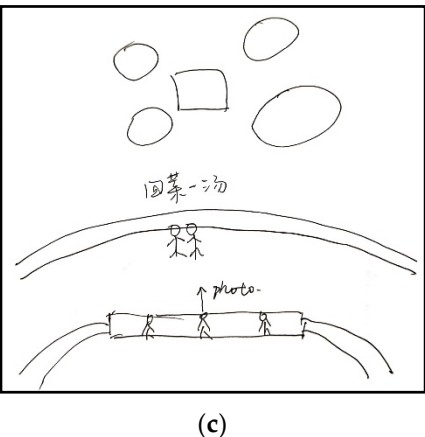

(**a**)                                        (**b**)                                        (**c**)

**Figure 3.** Subcategories of individual cognitive maps of tourists in traditional villages: (**a**) entity, (**b**) abstract, and (**c**) scene. Note: The Chinese letters in (**c**) is the name of Tulou buildings with "four dishes and one soup".

### 4.1.3. Hybrid Cognitive Map

The hybrid cognitive map not only highlights the landmark landscape, but also reflects the relationship between path and other spatial elements. The elements' role in landmark and path are quite similar, and neither is dominant. Compared with urban destinations, hybrid cognitive maps occupy a certain proportion of the total sample, and the number of traditional village destinations is only 20, accounting for 5.46%. This condition may be related to the spatial scale of the destination. The traditional village area is relatively small, and tourists have relatively low demand for identifying path elements. In particular, the prominent position of the main path in traditional villages is not as important as that of urban destinations, so the proportion of paths and landmarks drawn together on the cognitive map is low. The unique feature of the hybrid cognitive map found in this study lies in the evident spatial elements of traditional villages, that is, remote mountains around the traditional villages and landscape paths along rivers are considered as village boundaries (see Figure 4). This type of map reflects the gradual improvement of tourists' cognition of the overall spatial environment of traditional villages.

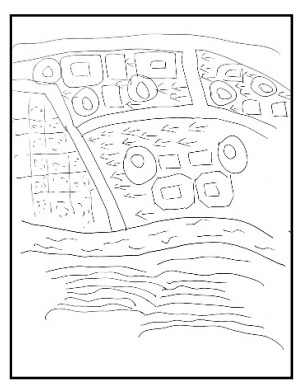 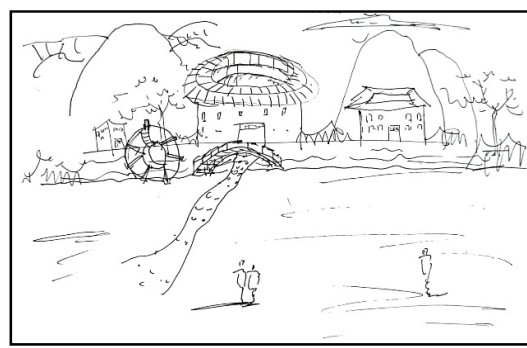 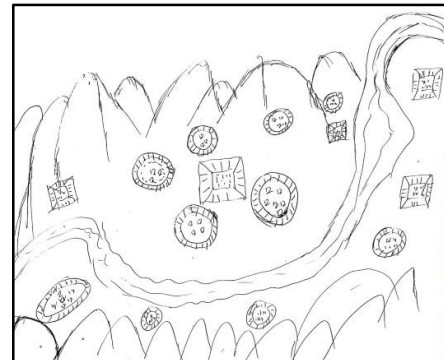

**Figure 4.** The hybrid cognitive map of tourists in traditional villages.

### 4.1.4. Sequential Cognitive Maps

The proportion of sequential cognitive maps is low at only 10, accounting for 2.73% of the total sample. They are divided into three subcategories.

(1)    Fragmented cognitive map (see Figure 5a). This type of cognitive map is mainly a fragment of a certain area of traditional villages outlined by tourists, mostly along village alleys or footpaths. Six maps are available, accounting for 1.64% of the total sample. Compared with the complexity of chain, branch/loop, and network [2],

fragment is the most elementary type, and more clearly reflects tourists' spatial cognition mode dominated by destination route.

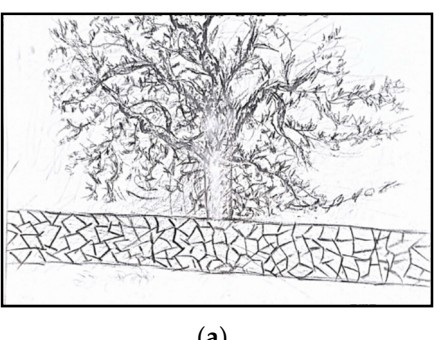 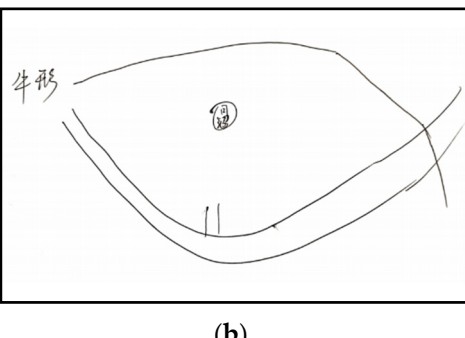 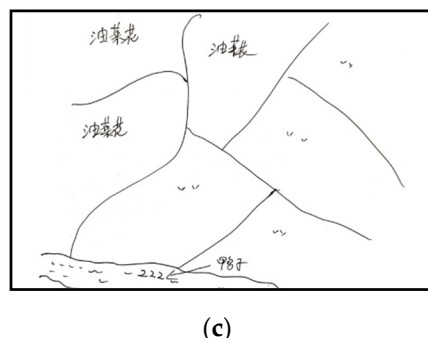

(**a**) (**b**) (**c**)

**Figure 5.** Subcategories of the sequential cognitive map of tourists in traditional villages: (**a**) fragmented, (**b**) branch/loop and (**c**) network. Note: The Chinese letters in (**b**) is the shape of an ox, and in (**c**) are rape flowers and ducks.

(2) Branch/loop cognitive map (see Figure 5b). This type only has two cognitive maps, accounting for 0.55% of the total sample. Tourists form cognitive maps according to the direction of traditional village branches/loop routes.
(3) Network cognitive map (see Figure 5c). Only two cognitive maps are available, accounting for 0.55% of the total sample. Tourists construct their overall cognition of traditional villages by connecting the traffic route network and regional functional zoning to form cognitive map.

*4.2. Analysis of Spatial Cognitive Elements*

The five basic elements of destination spatial image cognition overlap and echo each other, and jointly outline the most beautiful traditional village picture in the mind of tourists. Although the effects and intensity of each element in each cognitive map are different, the cognitive map features of tourists with the same attributes should be basically stable, thereby reflecting their spatial image cognition rules.

According to the statistics (see Table 2), 366 cognitive maps of traditional villages drawn by tourists show the 5 basic elements. In terms of the occurrence frequency of elements, landmark (95.63%), path (42.08%), district (7.10%), edge (4.37%), and node (4.37%) were ranked from high to low. Evidently, landmarks play an absolutely dominant role in tourists' cognition maps of traditional villages, which is related to tourists' tourism motivation. As the core attraction of traditional villages, landmark landscapes are the most likely to leave profound spatial images on tourists. In addition to the high frequency of path elements, the frequency of districts, edges, and nodes is low; it is related to the scale range and geographical characteristics of traditional village destinations. The research results show the unique differences between traditional villages and urban destinations, indicating the value of this study to enrich the knowledge of spatial image in different destination types.

The appearance frequency and presentation of specific elements reflect the imageability of destination spatial cognition elements, indicating an important clue to explore the cognition rules of tourist spatial image [43]. The following is a specific analysis of spatial cognitive elements of traditional villages based on these two indicators.

**Table 2.** Cognitive elements of spatial image in traditional villages.

| Cognitive Elements | Occurrence Number | Occurrence Frequency (%) | Specific Elements |
|---|---|---|---|
| landmark | 350 | 95.63 | "Four dishes and one soup" Tulou building in Tianluokeng, Zhencheng building, Yuchang building, Hegui building, Hakka folk culture museum, moon pond, memorial archway of Hu Wenguang governor, grinding time shop, Half-sugar coffee house, Kunlun youth hostel |
| path | 154 | 42.08 | Alleys, riverside footpath, and expressways outside traditional villages |
| animals and plants | 146 | 39.89 | Distant mountains, green plants, ancient trees, rape flowers, vegetable gardens; ducks, dogs, birds |
| district | 26 | 7.10 | Tianluokeng and Hongkeng building group of Fujian Tulou, Xidi and Hongcun building group of southern Anhui |
| edge | 16 | 4.37 | Mountains, the rivers |
| node | 16 | 4.37 | Small bridge |

### 4.2.1. Landmark

Landmarks are the most frequent among all elements, indicating that tourists mainly rely on landmarks of traditional villages to form their destination spatial cognitive. The "Four dishes and one soup" Tulou building in Tianluokeng, the Zhencheng building, the Yuchang building, the Hegui building, the moon pond, the memorial archway of the Hu Wenguang governor, the grinding time shop, and other specific elements are the most symbolic landmark in the minds of tourists. Results show that in addition to traditional cultural landmarks, such as Tulou buildings, the moon pond and memorial archways, and activation landmarks, such as shops, hostels and cultural pavilions, rely on traditional dwellings. Therefore, iconic buildings and cultural landscapes that can represent the image of traditional villages are important references for tourists' cognition of spatial image.

### 4.2.2. Path

Path is the second most frequent among all elements, indicating that traditional village path has high imageability. The path element appears in the spatial cognitive map, which reflects the cognitive role of tourists in constructing the spatial image of traditional villages by means of path. The paths of traditional villages are different from the main paths of urban destinations, and more of them are alleys, riverside footpaths, and expressways outside traditional villages. Although these paths are "nameless", they form an important part of the traditional village tourism experience along with the landmark landscape of the traditional village. Tourists stroll through the alleys of the traditional village with the sound of flowing water, enjoying the beauty of rural life.

### 4.2.3. District

The district in this study is reflected in four architectural groups formed by multiple residential buildings, such as Tianluokeng and Hongkeng of the Fujian Tulou building group, the Xidi and the Hongcun building group of traditional villages in southern Anhui. Although the frequency of district element is not as high as that of single landmark buildings, the traditional village landscape and architectural techniques experienced by tourists are impressive, thereby strengthening the ability of district cognition in destination spatial image.

### 4.2.4. Edge

Edges are generally accompanied by districts. Therefore, the mountains around traditional villages and the rivers that flow through them become cognitive elements

to divide districts. These edge elements are mainly artistic edges in tourists' cognitive maps, without specific reference. In terms of occurrence frequency, the frequency of edge elements is relatively low, which is related to the fact that tourists focus more on individual landmarks rather than on the whole district.

### 4.2.5. Node

Node elements appear less frequently and are the worst cognitive elements in tourists' mind, similar to edge elements. The node elements in this study are only reflected in the small bridge, a specific element, which is related to the position of the small bridge connecting the main tour route of traditional villages. This element becomes an important reference point for tourists to recognise the spatial structure of traditional villages.

### 4.2.6. Animals and Plants

Although animals and plants are not the five basic elements of spatial image proposed by Lynch, they cannot be ignored in tourists' spatial cognitive map in the special destination context of "harmonious coexistence between man and nature". According to the statistical data of this study, the frequency of occurrence of animals and plants is 146, accounting for 39.89% of the total sample, only lower than the 2 elements of landmark and path, and remarkably higher than the 3 elements of district, edge, and node. The elements composed of distant mountains, green plants, ancient trees, rape flowers, vegetable gardens, and animals, such as ducks, dogs, and birds, are not only the cultural landscape of traditional villages, but also the life scenes above the landscape, which occupy a "special" weight in tourists' cognition of spatial image. Therefore, this study will be added as a basic cognitive element of traditional villages.

### 4.3. *Analysis of Spatial Image Cognitive Process*
### 4.3.1. Cognitive Process of Map Types

According to tourists' stay time in traditional villages, the evolution trend of spatial cognitive map types with tourists' stay time is studied to confirm the tourists' cognition process of a traditional village spatial image. The change trend figure of the cognitive map in the length of stay days of tourists in traditional villages (see Figure 6) shows that the distribution of the four cognitive maps in different stay days is not balanced; spatial and individual types have always been the main cognitive map types of tourists in traditional villages. From the perspective of cognitive map type change, spatial cognitive map always occupies the highest position, and the proportion of stay days of four days and above is the same as that of the individual map. The individual cognitive map is always in the second place in proportion. The hybrid cognitive map is always in the third place in proportion, and the proportion of stay days of the third day is the same as the sequential type. The sequential cognitive map is always in the lowest proportion. Therefore, the tourists' cognitive map type of spatial image of traditional villages is stable, and no conversion trend amongst different cognitive map types is observed. The perspective of the change trend of each type indicates the three characteristics of change. Firstly, spatial and hybrid type showed a trend of "low → high → low", reaching the highest proportion of each type on the second day, and then decreasing with the increase in stay days. This finding is in line with the rules of destination spatial cognition. Spatial and hybrid types are relatively complex, and tourists can establish a complete cognitive spatial image by familiarizing themselves with the destination tour in the first two days. When the journey is coming to an end, tourists will remember the most impressive landmark landscape and consider it an important reference for the spatial image of traditional villages, thereby promoting a certain advantage in preserving the proportion of the individual cognitive map. Secondly, the individual type shows a trend of continuous decline. For tourists on the first day, they are generally impressed by the landmark landscape they have visited, which is the direct reason for the largest proportion of individuals on the first day. Thirdly, the

overall number of sequential types is small, showing a trend of small fluctuations with the length of stay days.

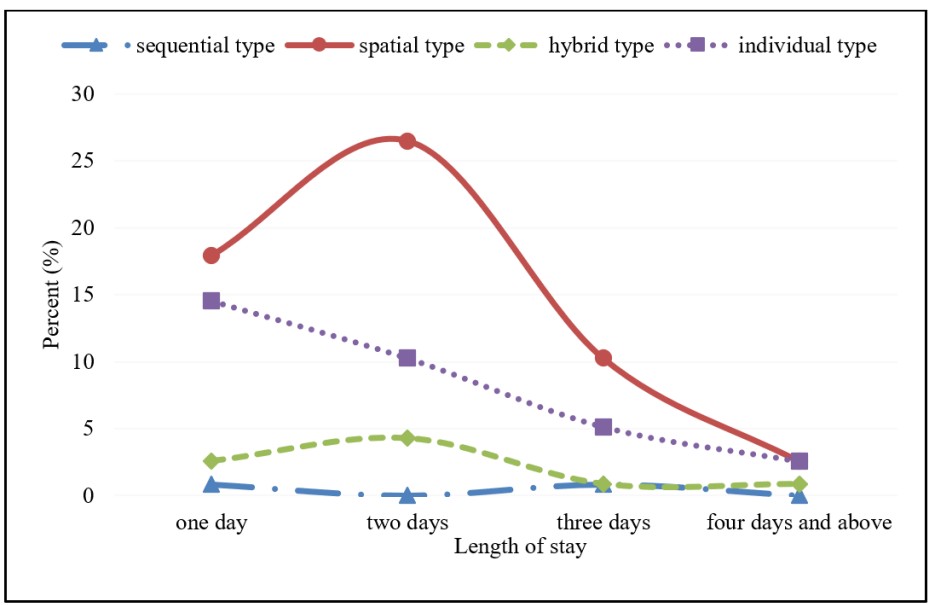

**Figure 6.** Evolution trend of four cognitive maps in terms of tourist stay days.

4.3.2. Cognitive Process of Spatial Elements

The evolution rule of staying time of spatial cognitive elements in traditional villages is also an important aspect of exploring the cognitive process of the spatial image. The change trend figure of the cognitive element in the length of stay days of tourists in traditional villages (see Figure 7) indicates that the distribution of the six cognitive elements in different stay days is not balanced; landmarks, paths, and animals and plants have always been the main cognitive map elements of tourists in traditional villages. From the perspective of cognitive element type change, landmarks are always in the highest proportion. The order of paths, and animal and plant elements, changed alternately. Path is in the second place in the first two days, whereas animal and plant elements rose to the second place in the third day and subsequently. For district elements, except the first day, the three other stay days always ranked fourth. The number of edge and node elements is small. Except for the first day, the three other stay days are in the alternate fifth and sixth positions. Therefore, tourists' spatial cognition element types of traditional villages have the characteristics of slight fluctuation. The perspective of the change trend of each type indicates two characteristics of change. Firstly, landmark, path, animal and plant, and district showed a trend of "low → high → low", reaching the highest proportion of each type on the second day, and then decreasing with the increase in stay days. This finding is in line with the rules of destination spatial cognition. By visiting traditional villages in the first two days, tourists can establish a relatively complete spatial image of the four elements of landmark, path, animal and plant, and district. When the journey is coming to an end, the landmark of traditional villages will leave a deep impression on tourists and generate good memories. Thus, the landmark remains the cognitive element with the highest frequency in the last days of stay. Secondly, the overall number of edge and node elements is small, showing a trend of small fluctuation and decline. Tourists on the first day generally need to find their way and so focus more on the edges and nodes of traditional villages. With the deepening of familiarity, the attention to edge and node elements decreases.

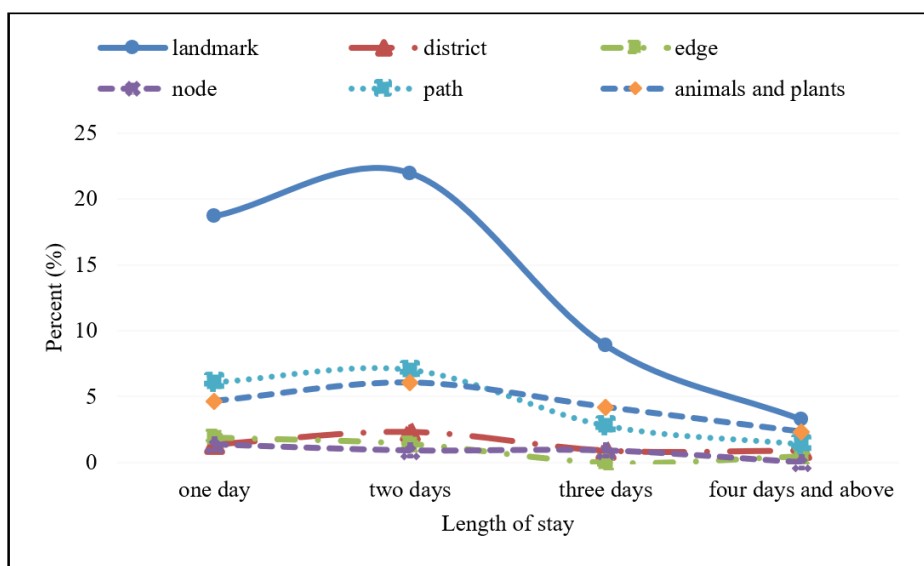

**Figure 7.** Evolution trend of six cognitive elements in terms of tourist stay days.

## 5. Discussion and Conclusions

This study explores the spatial image cognition rules of traditional villages from the perspective of tourists' hand-drawn sketches, and enriches the application of the cognitive map method in the tourist experience of traditional villages. The results show that spatial image cognitive maps were divided into four types. The spatial type was dominant, followed by individual, hybrid, and sequential types. The spatial image cognitive elements were divided into six types, of which the frequency of landmark was the highest, followed by path, and animals and plants. The frequency of district, edge, and node was low. On this basis, we further studied the evolution trend of the spatial image cognitive map and elements in relation to tourists' stay days. In terms of dominant types of cognitive map, the evolution sequence of "spatial + individual → spatial + individual + hybrid → spatial + individual" was presented. In terms of the dominant elements of spatial cognition, a developmental sequence of "landmark + path + animal and plant → landmark + animal and plant + path" was found. This spatial cognitive process reflects the transformation of tourists' focus on the spatial image of traditional villages from individuality to structure and then to meaning.

### 5.1. Theoretical Implications

This study presents four theoretical contributions. Firstly, spatial cognitive map has an absolute advantage in tourists' spatial cognition of traditional villages. Previous studies on urban residents found that, except for Huynh et al. [28], sequential cognitive maps mostly accounted for about 70% of responses, with evident advantages, and that the proportion of spatial maps with other types was less than 30%. However, in the study of the spatial image of tourist destinations, the proportion of the spatial cognitive map of domestic tourists is nearly 40%. Taking non-urban destinations as case studies, results of a study in Finnish Lapland showed that the proportion of spatial and sequential maps are almost equal or higher [11]. This study also shows that the spatial type has a proportional advantage in tourists' cognitive map. These differences show that the two groups have different ways of constructing spatial images, fundamentally caused by the differences in behaviour patterns. For residents with a large range of activities and abundant travel routes, their cognitive map reflects a wider range of cities [2]. The accumulation of urban spatial environment knowledge comes from various travel routes experienced by residents in their life [44]. Therefore, this is the main reason for the high proportion of sequential types in residents' cognitive map. Tourists' spatial cognition of destinations is a functional cognition driven by tourism demand and a process of actively acquiring and recognising spatial information due to tourism [15]. Residential buildings, ancestral halls, and other spatial elements

reflecting the characteristics of traditional villages are not only the main clues for tourists to locate, but also the districts where their personal experience is concentrated. These buildings are scattered in every corner of traditional villages, directly leading tourists to depict the whole space of traditional villages with patterned cognitive maps. Therefore, the spatial cognitive map has more advantages in traditional village spatial cognition.

Secondly, the landmark is the dominant element of tourists' spatial cognition of traditional villages. The number of landmarks is the largest factor in the initial period of tourists' stay, and it always occupies the first position, meaning that it is undoubtedly the dominant element of tourists' spatial cognition of traditional villages. Most landmarks are closely related to tourism attractions, indicating that tourists focus on selecting important information conducive to solving tourism problems and, thus, form an understanding of the spatial environment of traditional villages. Lynch pointed out that tourists initially rely on landmarks as a positioning basis when going to a place, and then gradually develop a more detailed cognitive map containing path elements [1]. As reference points in spatial cognition, landmarks are an important basis for people's positioning in unfamiliar environments [23]. They are the initial anchor point of spatial cognition, and the image of path structure is gradually enriched and developed after the formation of a landmark network [45]. The results of this study support these assertions and conform to the anchor point theory proposed by Golledge [46]. Representative dwellings in traditional villages are the most frequent feature in the landmark elements and are the most important initial anchor point for tourists to recognise traditional villages, and the connection between these anchor points is the alley and riverside footpaths. The appearance of important elements in the cognitive map reflects the cognitive sequence that tourists pay attention to landmarks initially, and then connect them to form path structure images. Traditional villages are not only objectively existing villages, but also subjectively constructed by visiting tourists. The tourism experience value of traditional villages obtained by tourists depends on the symbolic cultural significance of the landmark scenic spots they choose [47]. Therefore, in addition to spatial positioning and way-finding, the pursuit of maximising the value of the tourism experience, as well as the emotional connection with the destination, promote landmarks to become the dominant elements of tourists' spatial cognition.

Thirdly, destination environmental characteristics constitute the main elements of tourists' spatial cognitive map. Geo-spatial cognition is influenced by environmental characteristics and individual cognition [48]. For the same group, the difference in urban spatial morphological characteristics often leads to the difference in dominant elements and types of cognitive map. Lynch [1] found that residents of cities with different spatial structures construct their cognitive maps with different elements. Boston residents often describe the city spatially in terms of districts, Jersey City residents rely more on main paths and Manhattan Island, and Los Angeles residents tend to describe a grid of path systems. Madrid tourists use spatial knowledge of maps to personalize the city's travel maps. Participants can be divided into the following three types of tourists: "guided", "explorer", and "conditioned" [49]. Using mental mapping methods, Gieseking [50] identified the lake, library, entry gate, clock tower, and founder's grave as the main landmarks of a women's university in Massachusetts. Subsequent empirical studies have also found that the actual spatial form of a city has an impact on residents' spatial cognitive characteristics [2,34,35]. This study found that in the cognitive elements of tourists in traditional villages, besides the traditional landmarks and paths, the new cognitive elements of animals and plants are in the third place, thereby reflecting that the living environment characteristics of tourism destinations constitute the main elements of tourists' spatial cognitive map. As a cultural destination, traditional villages not only have rich historical heritages, but also have the earthbound landscape and pastoral life that city tourists yearn for. Tourists enjoy a harmonious way of life between man and nature in traditional villages. Thus, the environmental characteristics, such as ancient trees, green plants, dogs, and birds strengthen the role of location elements in tourists' cognition of spatial image. Furthermore, the environmental characteristics of tourism destinations, especially the living atmosphere

above the landscape, are important moderating variables that affect the dominant elements of tourists' cognitive map.

Fourthly, the cognitive process of the spatial image of traditional villages reflects the transformation of tourists' cognitive emphasis on destination environmental image. Research on culture shock shows that people have the strongest psychological response to a new cultural environment, and when people stay longer in a new environment and become accustomed to it, they are likely to adjust their psychology to their state in a familiar environment [51]. The theory of the environment–behaviour relationship holds that the occurrence of behaviour is caused by the interaction between internal individual factors and external social environmental factors. People not only passively adapt to the environment, but also actively choose and use environmental factors. The ultimate goal is to pursue the dialectical unity of environment and behaviour, and to improve the quality of life of people [52]. The results show that tourists with different days of stay have different impressions of map types and different dominant elements of spatial cognition of traditional villages. Tourists who stay for a short period of time are deeply impressed by the landmarks of traditional villages, and they understand the spatial structure of traditional villages through auxiliary cognitive means, such as a tour guide's explanation, residents' conversation, tourism brochures, and map navigation, needed to arrange subsequent travel itineraries. Tourists who stay for a long time have direct experience of the landmarks and paths of traditional villages. With the further deepening of familiarity, they enjoy the slow life in a quiet environment, allowing their restless hearts to rest in the traditional villages. The main cognitive map types (spatial, individual, and hybrid) and dominant cognitive elements (landmark, path, animals and plants, and district) found in this study showed a "low → high → low" evolution trend. That is, the proportion of each type reaches the highest on the second day, and then decreases with the extension of stay time, consistent with the relevant research conclusions of culture shock theory [51,53]. This reflects the transformation of tourists' cognitive emphasis on destination environmental image.

### 5.2. Practical Implications

This study provides a reliable and valid method for destination managers, tourism planners, and tourism marketers to better understand tourists' spatial image cognition of traditional villages, and then puts forward policy suggestions for optimising the protection and development of traditional villages. Firstly, the landmarks in the tourists' hand-drawn sketches are traditional residential dwellings and activation cultural landscapes, reflecting the tourists' desire to find the ideal lifestyle of modern people from the wisdom of traditional human settlements. Therefore, destination managers should innovate the protection and utilisation of traditional architecture and actively explore the "traditional architecture +" integrated development model, such as traditional architecture and education, traditional architecture and studio, and traditional architecture and cultural square. The tourists are encouraged to closely understand the traditional village culture and to improve their happiness of travel life. Secondly, in the cognitive maps of the tourists' spatial image, the patterned subcategory of the spatial image has the highest proportion (53.55%). These cognitive maps are mostly centred on residential buildings, with surrounding mountains and rivers as the village boundaries, forming a living space pattern of poetic dwelling in traditional villages. Therefore, tourism planners of destinations should establish the development concept of life above landscape, and design sustainable development planning of traditional residential buildings, and village living environment as a whole to make tourists feel that traditional villages are their homes. Thirdly, tourism marketers should precisely market traditional village tourism products according to the needs of different tourists. For the mass tourists, while doing a good job in tourism reception services, we provide strong participation in folk customs experience and cultural performances to extend the stay time of tourists. For leisure tourists, the interaction between the host and the guest should be enhanced, and tourists should be invited to participate in the daily life of the villages to shorten the psychological distance between them and make them feel the hospitality of

the residents. For potential tourists, we should focus on the marketing of earthbound atmosphere and architectural culture to attract tourists with characteristics different from mass tourism destinations.

*5.3. Limitations and Future Research*

Several limitations of this study should be acknowledged. Firstly, this study is based on case studies of traditional villages in the Anhui and Fujian provinces of China. The universality of the research conclusions needs to be extended to traditional village destinations in other provinces. Secondly, this study is conducted from the perspective of tourists' hand-drawn sketches. However, destination spatial image cognition is a complex concept, and future research should collect data by combining scenario experiments, in-depth interviews, and questionnaire surveys to more comprehensively explore the rules of tourists' spatial image cognition. Thirdly, this study analyses the evolution process of spatial image cognition of tourists in traditional villages in terms of stay days. Future research can explore the differences of demographic variables in spatial image cognition.

**Author Contributions:** Research design, Z.J. and Y.S.; conceptualization, Y.S.; investigation, Y.S.; writing—original draft preparation, Z.J.; funding acquisition Z.J. All authors have read and agreed to the published version of the manuscript.

**Funding:** This work was supported by the Huaibei Education Science Research Project, China under grant number HBJK1709; Sichuan Cuisine Development Research Center Planning Project, Key Research Base of Sichuan Federation of Sciences Association, China under grant number CC17G08.

**Institutional Review Board Statement:** Not applicable.

**Informed Consent Statement:** Not applicable.

**Data Availability Statement:** Not applicable.

**Acknowledgments:** The authors thank the reviewers' beneficial suggestions that helped improve this paper.

**Conflicts of Interest:** The authors declare no conflict of interest.

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
