# Peer review of "Exploring the Spatial Image of Traditional Villages from the Tourists’ Hand-Drawn Sketches"

_sustainability, doi:10.3390/su14105977_

Round 1
Reviewer 1 Report
Dear Editorial Board, Dear Authors,
I found the choice of the research topic very interesting and timely current.
The paper entitled “Exploring the Spatial Image of Traditional Villages from the Tourists’ Handdrawn Sketches” analyses the spatial image characteristics of four traditional villages of World Cultural Heritage sites in China through the use of tourists’ hand-drawn sketches by a sample of 366 respondents and further explores the evolution process of cognitive map types and constituent elements with tourists’ stay days.
The paper asks the following research questions:
(1) What are the dominant elements of spatial image cognition of tourists in traditional vil- lage destinations?
(2) Is landmark dominant or path dominant?
(3) What is the evolution process of spatial image cognition of tourists in traditional villages?
(4) Does it show a pattern of transformation from sequential type to spatial type, consistent with urban destination?
After reading the paper, I have comments and suggestions to improve the paper as follows:
Layout and structure of the work are correct
Abstract - I suggest to improve the abstract according to the journal guidelines. Information about the applied research method is missing.
In the Introduction
The literature review is well laid out, but refers primarily to literature from China. No reference to other areas in the world with similar conflicts has been made. This study is pioneering with respect to China . However, the journal Sustainability has a global character and the research problem of the article should not only concern regional problems.
The Results are presented and described in a very good way and are very interesting. They contribute to the value of this study. The different phases of the research are very well presented and interesting.
In the Discussion and Conclusion
An important element of the work is the attention paid to the protection and development of traditional villages. In the conclusion, the authors rightly emphasize that more efforts should be put into the protection and use of traditional architecture and actively explore the "traditional architecture + " integrated development model, such as traditional architecture + education, traditional architecture + studio and traditional architecture + cultural square.
At the same time, they point out that tourism planners in tourist destinations should establish the concept of developing life beyond the landscape and develop sustainable planning for the development of traditional housing and the village living environment as a whole, so that residents feel that traditional villages are their homes.
The above mentioned conclusions apply to the study region. Meanwhile, the Journal is international in scope and should refer to other studies with similar themes. In the article the authors presented only one example Boston.
In the Discussion Section, the authors should compare their project and results with results from similar conducted research on this topic from around the world.
Kind regards,
Reviewer 2 Report
Dear author/s,
after I read the manuscript: "Exploring the Spatial Image of Traditional Villages from the Tourists’ Hand-drawn Sketches", I have a few observations and recommendations:
- The objectives and the research questions are clearly stated.
- In the methodology section I was expecting to see more information about the tourism demand and offer from the research area (no of tourists, no of accommodation units, length of stay etc.).
- Please add a currency rate to USD or EURO, it will easier for international readers to understand and have an idea about the income level of the respondents.
- Please explain how the respondents were selected? Do you consider that the sample is representative? How was test the validity and reliability of the questionnaire?
- Please emphasize the link between the paper and the aim of the journal.
Good luck!
Round 2
Reviewer 2 Report
Dear author/s,
thank you for the improved version of the manuscript.